Prediction of epileptic seizures based on multivariate multiscale modified-distribution entropy

Aung Si Thu
Wongsawat Yodchanan yodchanan.won@mahidol.ac.th
Department of Biomedical Engineering, Faculty of Engineering, Mahidol University , Phuttamonthon, Nakhon Pathom , Thailand
Duraisamy Jude
Electronic publication date: 2021 Oct 15
Publication date: 2021
Volume: 7
Electronic Location ID: e744
Received 2021 Jul 21; Accepted 2021 Sep 22
Copyright: © 2021 Aung and Wongsawat
Copyright year: 2021
Copyright holder: Aung and Wongsawat
License: This is an open access article distributed under the terms of the Creative Commons Attribution License, which permits unrestricted use, distribution, reproduction and adaptation in any medium and for any purpose provided that it is properly attributed. For attribution, the original author(s), title, publication source (PeerJ Computer Science) and either DOI or URL of the article must be cited.
License URL: https://creativecommons.org/licenses/by/4.0/

Keywords: Distribution entropy, ANN, EEG, Entropy, Epilepsy

Funding: National Higher Education Science Research and Innovation Policy Council, PMU B (Project ID 25640001129328) This project is supported by the National Higher Education Science Research and Innovation Policy Council, PMU B (Project ID 25640001129328). There was no additional external funding received for this study. The funders had no role in study design, data collection and analysis, decision to publish, or preparation of the manuscript.

==============================
Epilepsy is a common neurological disease that affects a wide range of the world population and is not limited by age. Moreover, seizures can occur anytime and anywhere because of the sudden abnormal discharge of brain neurons, leading to malfunction. The seizures of approximately 30% of epilepsy patients cannot be treated with medicines or surgery; hence these patients would benefit from a seizure prediction system to live normal lives. Thus, a system that can predict a seizure before its onset could improve not only these patients’ social lives but also their safety. Numerous seizure prediction methods have already been proposed, but the performance measures of these methods are still inadequate for a complete prediction system. Here, a seizure prediction system is proposed by exploring the advantages of multivariate entropy, which can reflect the complexity of multivariate time series over multiple scales (frequencies), called multivariate multiscale modified-distribution entropy (MM-mDistEn), with an artificial neural network (ANN). The phase-space reconstruction and estimation of the probability density between vectors provide hidden complex information. The multivariate time series property of MM-mDistEn provides more understandable information within the multichannel data and makes it possible to predict of epilepsy. Moreover, the proposed method was tested with two different analyses: simulation data analysis proves that the proposed method has strong consistency over the different parameter selections, and the results from experimental data analysis showed that the proposed entropy combined with an ANN obtains performance measures of 98.66% accuracy, 91.82% sensitivity, 99.11% specificity, and 0.84 area under the curve (AUC) value. In addition, the seizure alarm system was applied as a postprocessing step for prediction purposes, and a false alarm rate of 0.014 per hour and an average prediction time of 26.73 min before seizure onset were achieved by the proposed method. Thus, the proposed entropy as a feature extraction method combined with an ANN can predict the ictal state of epilepsy, and the results show great potential for all epilepsy patients.

Introduction

Epilepsy is one of the most common neurological disorders of the nervous system, affecting approximately 50 million people worldwide, and approximately five million people are diagnosed with epilepsy each year (World Health Organization (WHO), 2019). Therefore, the social and economic impacts on patients with epilepsy are becoming increasingly concerning. Although the seizures of 70% of epileptic patients can be controlled by antiseizure medicines, the seizures of 30% of patients with epilepsy cannot be treated by either medicines or surgery; therefore, these patients must live their whole lives with epilepsy, and their seizures can occur anytime and anywhere (Fujiwara et al., 2015). Electroencephalogram (EEG) can record the patients’ brain activities and be used as a tool for diagnosing and analyzing epilepsy (Wang et al., 2010). Thirty percent of epileptic patients whose seizures cannot be controlled urgently need a system that can improve their lives, by successfully predicting a seizure before it begins. However, epilepsy prediction remains one of the competitive challenges for researchers, and numerous methods have already been proposed to address this problem. Scholars approach this issue in various ways, e.g., linear methods (Salant, Gath & Henriksen, 1998) and nonlinear dynamics (Iasemidis et al., 1990). For linear measurement, statistical measures, including the calculation of variance, skewness, and kurtosis, are used for several seizure prediction tools, and researchers have described that kurtosis increase but variance decrease during the state of preictal activity (Aarabi, Fazel-Rezai & Aghakhani, 2009). The mean phase coherence (MPC) (Mormann et al., 2003), Shannon entropy index (Rosenblum et al., 2000), and conditional probability index are the best nonlinear measures compared to other nonlinear features (Mormann et al., 2005). Moreover, the differential entropy with the cumulative sum (CUSUM) procedure has been applied to predict seizures and shows 87.5% sensitivity, a 0.28 per hour false prediction rate and a 25 min average prediction time (Zandi et al., 2009). In other studies, the permutation entropy (PE) method has been used to extract features, and combined with a support vector machine (SVM) classification method, 94% sensitivity, a 0.111 per hour false prediction rate, and 63.93 min of the average prediction time were shown (Yang et al., 2018).

There are different types of methods for measuring time series complexity, e.g., entropies (Coifman & Wickerhauser, 1992), fractal dimensions (Mashiah et al., 2008), and Lyapunov exponents (Rosenstein, Collins & De Luca, 1993). However, entropy calculation becomes more interesting in the neuroscience field because of the nonstationary features of the EEG signals. Entropy is a method that can be used to distinguish the regular, chaotic, and random behavior of a time series by measuring complexity (Paluš, 1998). Moreover, the use of entropy combined with a Monte Carlo tree search (MCTS) process is the most effective method to addredd the container loading problem (Cant et al., 2018); therefore, entropy is a method that can be used to measure disorder or irregularities in a wide range of applications (Howedi, Lotfi & Pourabdollah, 2020). Additionally, EEG signals from epileptic patients can be classified into three different states: interictal state, preictal state, and ictal state (see Fig. 1). The first state refers to the time between seizures, the second state is the time period just before the seizure arrives, and the last state is the seizure period (Chiang et al., 2011). In previous work, a new entropy method called modified-distribution entropy (mDistEn) was proposed, and this method successfully detects the different states of epileptic EEG signals by calculating the complexity of the signals (Aung & Wongsawat, 2020). Moreover, an effective coarse-grained calculation was added to the entropy method, which becomes multiscale modified-distribution entropy (M-mDistEn) (Aung & Wongsawat, 2021). The purpose of this multiscale method is to detect the various scales (frequencies) of the EEG signals; therefore, this method is usable for detecting motion artifacts. The main difference between the common entropy and the multiscale entropy is the evaluation of time series coarse-grained entropy to quantify the interdependency between entropy and scales (Costa, Goldberger & Peng, 2002). However, there are some limitations to multiscale entropy because it is designed for scalar time series analysis, and it is not suitable for accurately reflecting the complexity of multivariate time series in complex systems (Zhang & Shang, 2019).

Figure 1 Epileptic EEG signals from subject no.1 (A) frontal area, (B) occipital area, and (C) different area including central, temporal, and parietal brain areas.

The advantages of multivariate entropy can overcome the shortcomings of multiscale entropy, including evaluating within-and cross-channel dependencies in multiple data channels, assessing of the underlying dynamical richness of multichannel observations, and more degrees of freedom in the analysis than those of standard multiscale entropy (Ahmed & Mandic, 2011). For the reasons outlined above, entropy can distinguish the different states of epileptic EEG signals, and therefore, the prediction of epilepsy is possible according to numerous experiments (Yang et al., 2018). By exploring the advantages of multivariate methods and the previous entropy methods (mDistEn and M-mDistEn), a new method called MM-mDistEn is proposed, and this new method provides the crucial features extracted from epileptic EEG signals and applies these features to ANNs (Siddique & Adeli, 2013) in seizure prediction systems. The proposed system also reveals improved results in all performance measures; thus, it may be another alternative method for helping epileptic patients predict seizures before they start.

Materials & methods

A detailed explanation of the calculation of the proposed entropy method, MM-mDistEn, is mentioned in this section. The classification of epilepsy is also performed by using an ANN, and the step-by-step procedure is described after the explanation of the parameter selection. In this paper, the public dataset is used for experimental data analysis and introduced in the next subsection.

Dataset description

The signals used for analysis in this paper are from the public PhysioNet Database (Goldberger et al., 2000; Shoeb, 2009b), where the CHB-MIT dataset of EEG signals with seizure events is used for data analysis. The dataset is freely downloadable via the link that is provided in the reference section (Shoeb, 2009a). Data were collected as previously described in (Daoud & Bayoumi, 2019) and which include long-term scalp EEG data form pediatric subjects with intractable seizures. Recordings, grouped into 23 cases, were collected from 22 subjects (five males, ages 3–22; and 17 females, ages 1.5–19), and these subjects were monitored for up to several days following withdrawal of antiseizure medication to characterize their seizures and evaluate their candidacy for surgical intervention (Shoeb, 2009b). Most files contain 23 EEG signals (24 or 26 in a few cases). The international 10–20 system of EEG electrode positions and the sampling rate is 256 per second with 16-bit resolution (Goldberger et al., 2000). Fig. 1 shows a 1-h recording of epileptic EEG signals from subject no.1. The different states of interictal, preictal, and ictal of EEG signals from the frontal area, occipital area, and different area of the brain are clearly shown in Fig. 1.

Multivariate multiscale modified-distribution entropy

A new method, MM-mDistEn, was used to calculate the data multidimensionally. Three steps are required to calculate the entropy values, as shown in Fig. 2.

Figure 2 Block diagram for calculating multivariate multiscale modified-distribution entropy.

The algorithm is calculated as follows:

Step 1. Multivariate time series

First, the multivariate time series from the given time series data is constructed. The new input multichannel EEG can be set and notated as xc,i where c is the number of channels (1, 2, …, C), and i is the number of samples in each channel (1, 2, …, N).

Step 2. Coarse-graining process

The coarse-grained time series can be calculated using the multivariate time series data according to the scale factor, and the equation can be expressed as

(1) gc,js=1s∑i=(j−1)s+1j⋅s⁡xc,i,(1≤j≤Ns),

where gc,js is the multivariate coarse-grained time series, s is a scale factor and Ns=Ns.

Step 3. Calculate MM-mDistEn

Phase-space reconstruction is performed before the calculation of the entropy values, and the reconstruction is as follows:

(2) Ms(j)=[g1,jsg1,j+τs…g1,j+(m−1)τsg2,jsg2,j+τs…g2,j+(m−1)τs⋮gc,js⋮gc,j+τs⋮…⋮gc,j+(m−1)τs],(1≤j≤Ns),

where m is the embedding dimension and τ is the time delay. For the current study, m = 3 and τ = 1 are used (more information available in the parameter selection subsection below).

The proposed method, MM-mDistEn, which is implemented based on distribution entropy, adds two more threshold parameters, ‘r’ and ‘n’, to existing parameters. r is set by multiplying the standard deviation of all data values by 0.2, and n is set to 2 (Aung & Wongsawat, 2020). For a given multivariate coarse-grained time series, Create matrix Xs(j) in term of Ms(j)by

(3) Xsj=[Ms(j),Ms(j+1),…,Ms(j+(m−1)τ)],(1≤j≤Ns−(m−1))

The distance matrix Dijs is computed as D^ijs divided by r and then squared (n = 2):

(4) Dijs=(D^ijsr)n.

A matrix D^ijs between Xs(i) and Xs(j) (1 ≤ i, j ≤ Ns – ( m−1), i ≠ j) is computed using the Euclidean method.

After obtaining Dijs, the empirical probability density function (ePDF) is calculated using the histogram approach from Dijs of the previous steps with the bin number, B. The probability for that number can be given as Pt, where t = 1, 2, 3, …, B.

MM-mDistEn can be calculated by the following equation with the distance matrix Dijs as follows:

(5) MM-mDistEn(m,τ,r,n,B,s)=-1log2(B)∑t=1BPt(Dijs)log2[Pt(Dijs)],(1≤i,j≤Ns-(m-1),i≠j).

Parameter selection

MM-mDistEn uses predefined values for selecting parameters. There are six parameters that are required to compute the entropy values. First, the time delay, τ and dimension, m are used for the reconstructing of phase-space, with values of 1 and 3, respectively (Li et al., 2015b). Next, the distance matrix, Dijs is calculated using the parameters r and n, where r is the tolerance number and n is the order of the function, and both parameters are applied in calculating the proposed entropy. A large r and n can lead to noise influence, whereas a small r and n can cause information lose; and therefore, the parameter r is equal to the standard deviation of the series multiplied by 0.2, and n is set to 2 (Aung & Wongsawat, 2020; Chen et al., 2007). When the ePDF is calculated, another parameter value called the bin number, B, is needed, and B is set to 64 for this estimation (Li et al., 2016). The scale factor, s, is also needed for calculating of the multivariate multiscale entropy values, and the scale values used in data analysis range from 1 to 15 (Acharya et al., 2015). Additionally, the different values are selected for these parameters and shown in the figures in results section.

Classification of epileptic seizures from the extracted features

In this paper, a multilayer perceptron (MLP), which is an ANN, is used for training and testing the data. First, feature calculation is performed by using the MM-mDistEn method before these features are input to the ANN. After that, the neural networks are implemented by using the library support from TensorFlow. These features are imported into the environment for calculation and separated into input data and target data, and then these data are split into two sets: preictal period for the test set and interictal period for training. The 12 units for the first hidden layer and second hidden layer are used in the neural networks. The model was trained with backpropagation and optimized with the RMSprop algorithm (Daoud & Bayoumi, 2019). The loss function used in this model is the binary cross entropy (Daoud & Bayoumi, 2019). The rectified linear unit (ReLU) activation function (Hahnloser et al., 2000) is used for the hidden layers to add nonlinearity and make strong robustness to clear the noise from the input data. The softmax activation function is selected for the output layer to classify the multiclass outputs, and interictal, preictal, and ictal states of the epileptic EEG signals (Usman, Khalid & Aslam, 2020). The networks for each patient are trained individually for all 24 subjects. Finally, the results are shown in the results section.

Seizure alarm system

The calculation of mean values is performed when the predicted features (PF) are generated from the ANN, and those values are used as the decision-making process of the seizure alarm system. The seizure features (SF) are selected from the duration of the ictal period of the EEG signals, and the mean values of those periods are calculated. Then, the mean values of SF and PF are used for comparison, i.e., if the former values are greater than or equal to the latter values, the alarm signal is triggered for upcoming seizure events. The flow chart for the seizure alarm system is shown in Fig. 3.

Figure 3 Flow chart algorithm for the seizure alarm system.

Results

In this section, two different analyses are conducted with two different datasets. The simulation data are used to test the consistency of the proposed entropy by changing the different parameters values. The experimental dataset is required for the next process of classification and epilepsy prediction. A detailed explanation of the results from these two analyses is described in the following subsections.

Analysis of simulation signals

The proposed entropy method is analyzed with two different datasets: simulation data and experimental data. For the first dataset, three different signal types; sine wave (50 Hz frequency), chaotic series, and Gaussian series are used (Li et al., 2015a), and the length of each signal is 400 samples. After that, these three series are simulated as a function of the scale factor with embedding dimensions and shown in Fig. 4. In Figs. 4A and 4C, these three series are plotted according to their entropy values, but the chaotic series and Gaussian series overlap with each other in embedding parameters of values 2 and 4. However, the entropy values of three different series plot well-defined over all scale factors (1 to 15) with an embedding parameter value of m = 3 and are visualized in Fig. 4B. Therefore, the value of 3 for the embedding parameter is chosen, and the detailed explanation of the parameters used in the calculation of MM-mDistEn is described in the parameter selection section. Although the dependence of MM-mDistEn is on the bin number, all three different series have upward trends regarding the bin values, B (20 to 29), and distinguish these three series, as shown in Fig. 5A. The time delay, τ values range from 1 to 10, and the MM-mDistEn values decrease with increasing parameter values (see Fig. 5B). In Fig. 5C, the tolerance numbers, r used in the entropy method are multiplied by the standard deviation of the series, and these values range from 0.05 to 0.9 with an increase of 0.05. The plotted MM-mDistEn values are chaotic, Gaussian and sine waves on the order of higher to lower values.

Figure 4 The values of entropies (unitless) on simulation data as a function of the scale factor with different embedding dimensions (A) m = 2, (B) m = 3, and (C) m = 4.

Figure 5 The entropy values (unitless) on simulation data with respect to the: (A) number of bins (20 to 29), (B) time delay (1 to 10), and (C) tolerance (0.05 to 0.9 multiplied by the standard deviation of the simulation data).

Analysis of the experimental data

The values of MM-mDistEn are plotted with the scale factors (1 to 15) and are shown in Fig. 6. Instead of plotting all datasets, the entropy values of epileptic EEG signals from four subjects are plotted, and the entropy values are different in order of the highest values for the ictal state and the lowest values for the preictal state (see Figs. 6A, 6C, and 6D). The highest complexity is for ictal EEG signals, which are deterministic chaotic dynamics, compared with normal EEG signals, which are stochastic dynamics (Li et al., 2015b). Although a different order is seen in Fig. 6B, MM-mDistEn can still distinguish the three states of the epileptic EEG signals.

Figure 6 The values of MM-mDistEn (unitless) in the experimental dataset with different subjects (A) subject no.1, (B) subject no. 4, (C) subject no. 7, and (D) subject no. 8.

The performance measure is performed by calculating the accuracy, sensitivity, specificity, and AUC (Li et al., 2018) for all the subjects from the CHB-MIT dataset.

(6) Accuracy=TP+TNTP+FP+TN+FN,

(7) Sensitivity=TPTP+FN,

(8) Specificity=TNTN+FP,

where TP and TN are the number of true positives and the number of true negatives, i.e., the classifier correctly labels the actual number of ictal and normal EEG signals and FP and FN are the number of false positive and false negatives, these two values indicate the number of ictal and normal signals that are incorrectly categorized by the classifier (Li et al., 2018). It is clearly shown that the proposed extraction method combined with an ANN achieves an average performance measure of 98.66% accuracy, 91.82% sensitivity, 99.11% specificity, and an AUC value of 0.84 (see Table 1). Moreover, the minimum scores of the performance measure are still effective with an accuracy of 95.2%, a sensitivity of 83%, a specificity of 93.67%, and an AUC value of 0.75 (among all 24 subjects, in Fig. 7).

Figure 7 Performance measures for the analysis of EEG signals with an ANN.

Table 1 Performance comparison between different prediction methods and our proposed method.

Feature
extraction method	Feature classification method	Accuracy (%)	Sensitivity (%)	Specificity (%)	AUC	False alarm rate (h−1)	Prediction rate (%) and time (min)	Training time (min)	References	
MM-mDistEn	ANN	98.66	91.82	99.11	0.84	0.014	99/26.73	3.5	–	
M-mDistEn	ANN	88	85	90	0.8	0.081	90/23	3.7	–	
PE	ANN	90	83	93	0.76	0.262	70/22	3.5	–	
N/A	MLP	83.63	84.67	82.60	N/A	0.174	N/A	7.3	(Daoud & Bayoumi, 2019)	
DCNN	MLP	95.41	92.8	94.1	N/A	0.072	N/A	12.5	(Daoud & Bayoumi, 2019)	
CNN	SVM	N/A	92.7	90.8	N/A	N/A	-/21	N/A	(Usman, Khalid & Aslam, 2020)	
N/A	CNN	99.3	N/A	99.6	N/A	0.5	N/A	N/A	(Gómez et al., 2020)	

Performance measures for predicting epileptic EEG signals

The performance of the proposed prediction algorithm is calculated based on three factors: the false alarm rate ( Rfa), the prediction time, and the prediction rate ( Rp) (Aarabi & He, 2014):

(9) Rfa=NfaHt,

(10) Rp=NpNt,

(11) Tavg=TpNp,

where Nfa is the number of false alarms, Ht is the total number of hours, Np is the number of predicted seizures, Nt is the total number of seizures, Tavg is the average prediction time and Tp is the prediction time. The performance values of all cases of the 24 subjects are shown in Fig. 8. The prediction time is defined as the time between the instant at which a seizure can be predicted, and the actual beginning of the seizure and 1-h (3,600 s) long epileptic EEG signals are used for the prediction system (see Fig. 9). A Tavg of 26.73 min is achieved among all cases from the 24 subjects. The proposed method with an ANN achieves an average false alarm rate of under 0.25 per hour, an average prediction rate of over 70%, and an average training time of fewer than 3.5 min (see Fig. 8).

Figure 8 Performance evaluation for the prediction of EEG signals with MM-mDistEn and an ANN.

Figure 9 Prediction time and false alarm based on the different states of epileptic EEG signals.

Discussion

According to previous works (Aung & Wongsawat, 2020), there are some limitations on multivariate time series analysis, and therefore, MM-mDistEn is proposed to overcome these limitations. First, the phase-space reconstruction and estimation of the probability density between vectors provide hidden complex information. The multivariate time series property of MM-mDistEn gives us more understandable information within the multichannel data. Moreover, the data are also analyzed at different scales (frequencies) so that insight information can be seen in different scales ranges. According to the above results section, the proposed method was tested with two different analyses: simulation data and experimental data. Testing with the simulation data is used if the information of the proposed entropy has strong consistency and less dependency on preset parameters. Regarding the experimental data, the different performance measures are also provided on the proposed entropy combined with an ANN to classify the three states of the seizure from epileptic EEG signals. Additionally, postprocessing of the seizure alarm system helps patients predict upcoming seizures before they occur. The different performance measures on the proposed MM-mDistEn are illustrated in Fig. 7, and the performance evaluation for the prediction of epileptic EEG signals is described in Fig. 8.

A summary of the performance comparison between the existing prediction methods that have used the same dataset and the proposed method is shown in Table 1. The proposed method obtains a higher accuracy score among these methods, excluding the method using a convolutional neural network (CNN) (Gómez et al., 2020), while the training time is shorter than that of the other methods such as M-mDistEn with an ANN, PE with an ANN, a deep convolutional neural network (DCNN) with MLP and MLP (Daoud & Bayoumi, 2019). The sensitivity of the proposed method obtains a better score than other methods, but it is slightly lower than DCNN with MLP and CNN with SVM. Although the specificity of the proposed entropy is marginally lower than that of the method using CNN, the best rate for false alarms is obtained by the proposed method. The false alarm rate is also crucial for the prediction of epilepsy, and it is the smallest rate among these methods. An important factor in the prediction of epilepsy is the prediction time because it enables the delivery of warning signals to patients in a timely manner. The proposed combined system of MM-mDistEn and an ANN can send an alarm on average 26.73 min before the actual seizure starts according to the results from the experiments in all 24 subjects; therefore, the prediction time of the proposed method is earlier than that with the method using a CNN with SVM (Usman, Khalid & Aslam, 2020).

Conclusions

In this paper, a new feature extraction method, called MM-mDistEn, was proposed for predicting of seizures through combination with an ANN. The proposed method efficiently explores the information from multiple variables with multiple time scales and analyzes the complexity of that time series. Two different analyses were performed: a simulation dataset is used to prove the existence of consistency, and an experimental dataset is applied to distinguish of the different states of epileptic EEG signals. The performance measures of the proposed method were provided for the classification of the interictal, preictal, and ictal states. The advantages of multivariate robust entropy provide an efficient method for extracting features from multichannel EEG recordings. Moreover, the seizure alarm system was added as postprocessing step, which can warn patients about an oncoming seizure before its onset by providing an adequate prediction of the time between the preictal and ictal states. The proposed combination method will only require an EEG acquisition system for real-time usage, and it can become useful not only for clinical applications but also for usage outside of the hospital for epilepsy patients. Therefore, a portable version for seizure prediction can become a reality by using the proposed method. Future studies are needed for real-time applications to detect more complex behaviors from the different EEG datasets.

Supplemental Information

Supplemental Information 1 Source Code of the Multivariate Multiscale Modified-Distribution Entropy and the Analysis Tools.

Click here for additional data file.

Additional Information and Declarations

Competing Interests

Author Contributions

Data Availability

The authors declare that they have no competing interests.

Si Thu Aung conceived and designed the experiments, performed the experiments, analyzed the data, performed the computation work, prepared figures and/or tables, authored or reviewed drafts of the paper, and approved the final draft.

Yodchanan Wongsawat conceived and designed the experiments, analyzed the data, performed the computation work, authored or reviewed drafts of the paper, and approved the final draft.

The following information was supplied regarding data availability:

Data can be found on PhysioNet: https://www.physionet.org/content/chbmit/1.0.0/.

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
