# Peer review of "Prediction of epileptic seizures based on multivariate multiscale modified-distribution entropy"

_PeerJ Computer Science, doi:10.7717/peerj-cs.744_

## Round 0.1 · original submission · Major Revisions

Please revise and resubmit. The reviewers have raised concerns in the implementation part, so please be sure to focus on that aspect

·

Basic reporting

Give the abbreviation for all the terms.

Contribution and future work to be detailed.

Check the modules of Figures 2 and 3.

Include recent papers in the references.

Experimental design

Table 1 . Comparison between different methods - For the top 3 references not given, the first one is the proposed work but the second(M-mDistEn) is published, and the accuracy mentioned in this table mismatch with the real paper. Also, give the reference for the PE.

Figure 3, Seizure alarm system - Features extraction using module should be MM-mDistEN - Check and give the optimized flow diagram.

Validity of the findings

Justify - Performance measures given in the recent paper is higher

Gómez, C., Arbeláez, P., Navarrete, M., Alvarado-Rojas, C., Le van Quyen, M., & Valderrama, M. (2020). Automatic seizure detection based on imaged-EEG signals through fully convolutional networks. Scientific reports, 10(1), 1-13.

Reviewer 2 ·

Basic reporting

This paper seems interesting. The studies of this work is missing in literature survey.
The paper is well organized and The work is quite interesting and significant.
(weak accept). Technically Paper is satisfactory.

Experimental design

-It would be better if authors can do some more analysis with discussion.

Validity of the findings

- The abstract is not well written, author is suggested to detail more in abstract , and must mention the novelty of the work with key findings.

Additional comments

no comments

·

Basic reporting

This paper is addressing an important issue. The research question and why this research is conducted is very clear. However, the choice of the research method is unclear. There are various entropy measures and all of them perform differently. Depends on the choice of entropy the parameters chosen for a specific entropy, the results will vary. How this is justified? The results are not compared with any other techniques. A public data set is used, hence, it should b possible to compare the results with other prediction techniques.

Experimental design

As mentioned above, the project aim and objectives are very clear and the research question is a valid question. My concern is that the paper is not very rigorous. The literature review is rather short. For example, a comprehensive review of entropy measures are represented in "An entropy-based approach for anomaly detection in activities of daily living in the presence of a visitor" or application of Monte Carlo is described in "An entropy-guided Monte Carlo tree search approach for generating optimal container loading layouts.".
The paper should have a Related work section where a comprehensive review is presented.

Validity of the findings

The results are valid. Although, as mentioned above, some form of comparison ar required. The suggestion is to include some of the papers that used the same dataset and present their results briefly.

The selection of six parameters for the entropy measure must be expanded.

Additional comments

Remove all references to "we". All sentences should be in a passive format.
Line 82, Likewise .. I do not believe this is comparable. EEG information should b represented separately.
Line 112, provide a brief statement before section "Dataset description" is presented.
Line 167-168, the selection of six parameters must be expanded.
Line 167-175, USE: Line 167-168, ...First, time delay, \tau, and dimension, m, ... (use a comma instead of brackets to introduce the parameters.
Line 211, In Figure 5(a) and Fig 5(c) ...
Line 251, Tavg is not included in the definition.
Figure 2 is not readable.
Figure 3 is not readable.
Figure 4 does not add value and should be removed.

---

## Round 0.2 · accepted · Accept

It can be accepted now as the authors have made the revisions.

·

Basic reporting

OK

Experimental design

OK

Validity of the findings

Justify - Performance measures given in the recent paper is higher.
Gómez, C., Arbeláez, P., Navarrete, M., Alvarado-Rojas, C., Le van Quyen, M., & Valderrama, M. (2020). Automatic seizure detection based on imaged-EEG signals through fully convolutional networks. Scientific reports, 10(1), 1-13.

Still not convinced with the response stating time is less. for the mentioned Gomez paper in table 1, stated time as N/A. Try to give input based on further analysis.

Additional comments

OK

·

Basic reporting

I am satisfied that the authors have addressed all my comments and made appropriate corrections to the paper according to my earlier comments.

Experimental design

Please see my comments above.

Validity of the findings

Please see my comments above.

Additional comments

The paper has improved in the revised version and I am happy to recommend the publication of the paper.